# CircleGAN: Generative Adversarial Learning across Spherical Circles

**Woohyeon Shim**
POSTECH CiTE
wh.shim@postech.ac.kr

**Minsu Cho**
POSTECH CSE & GSAI
mscho@postech.ac.kr

## Abstract

We present a novel discriminator for GANs that improves realness and diversity of generated samples by learning a structured hypersphere embedding space using spherical circles. The proposed discriminator learns to populate realistic samples around the longest spherical circle, i.e., a great circle, while pushing unrealistic samples toward the poles perpendicular to the great circle. Since longer circles occupy larger area on the hypersphere, they encourage more diversity in representation learning, and vice versa. Discriminating samples based on their corresponding spherical circles can thus naturally induce diversity to generated samples. We also extend the proposed method for conditional settings with class labels by creating a hypersphere for each category and performing class-wise discrimination and update. In experiments, we validate the effectiveness for both unconditional and conditional generation on standard benchmarks, achieving the state of the art.

## 1   Introduction

Generative Adversarial Networks (GANs) [9] aim at learning to produce high-quality data samples, as observed from a target dataset. To this end, it trains a generator, which synthesizes samples, adversarially against a discriminator, which classifies the samples into real or fake ones; the discriminator's gradient in backpropagation guides the generator to improve the quality of the generated samples. As the gradient is difficult to stabilize, recent methods [2, 10, 18, 31] have suggested using the Lipschitz continuous space for the discriminator so that its gradient is bounded under some constant with respect to input. In a similar spirit, recent work [22] introduces a hypersphere as an embedding space, which enjoys its boundedness of distances between samples and also their gradients, showing its superior performance against the precedent models.

The most important qualities of the generated samples are realness and diversity. In conventional GAN frameworks including [22], the discriminator can be viewed as evaluating realness based on a prototype representation, i.e., the closer a generated sample is to the prototype of real samples, the more realistic it is evaluated as. The single prototype, however, may not be sufficient for capturing all modes of real samples, and thus recent work [1, 7, 19, 26] attempts to tackle this issue by employing multiple discriminators, i.e., multiple prototypes. But, training with multiple discriminators requires higher memory footprints and computation costs, and also introduces another hyperparameter, i.e., the number of discriminators. Conditional GANs [17, 21] use additional information of class labels to increase the coverage of modes across different categories. However, the class label does not ensure the intra-class diversity [21] while being beneficial to learn representative features of the class. Hence, the problem of mode collapse remains even in conditional models.

In this paper, we address the sample diversity issue by learning a structured hypersphere embedding space for the discriminator in GANs. Our discriminator learns to populate realistic samples around a great circle, which is the largest spherical circle, while pushing unrealistic samples toward the poles perpendicular to the great circle. In doing so, since the longer circles occupy the larger area on the

hypersphere, they encourage more diversity in representation learning of realistic samples. As the result, multiple modes of real data distribution are effectively represented to guide the generator in GAN training. We also extend the proposed method for conditional settings with class labels by creating a hypersphere for each category and performing class-wise sample discrimination and update. In experimental evaluation, we validate the effectiveness of our approach for both unconditional and conditional generation tasks on the standard benchmarks, including STL10, CIFAR10, CIFAR100, and TinyImagenet, achieving the state of the art.

## 2 Related Work

### 2.1 Generative Adversarial Networks

Previous work for improving GANs concentrates on addressing the difficulty of training. These studies have been conducted in different aspects such as network architectures [10, 13, 14, 23], objective functions [12, 20] and regularization techniques [2, 10, 18, 31], which impose the Lipschitz constraint to the discriminator. SphereGAN [22] has shown that using hypersphere as an embedding space affords the stability in GAN training by the boundedness of distances between samples and their gradients. Our work also adopts a hypersphere embedding space, but proposes a different strategy in structuring and learning the hypersphere, which will be discussed in details.

The most relevant line of GAN research to ours is on the lack of sample diversity. In many cases, GAN objectives can be satisfied with samples of a limited diversity, and no guarantee exists that a model in training reaches to generate diverse samples. To tackle the lack of diversity, a.k.a. mode collapse, several approaches are proposed from different perspectives. Chen et al. [5] and Karras et al. [14] modulate normalization layers using a noise vector that is transformed through a sequence of layers. Yang et al. [28] penalize the mode collapsing behavior directly to the generator by maximizing the gradient norm with respect to the noise vector. Yamaguchi and Koyama [27] regularize the discriminator to have local concavity on the support of the generator function to increase its entropy monotonically at every stage of the training. Liu et al. [16] propose a spectral regularization to prevent spectral collapse when training with the spectral normalization, which is shown to be closely linked to the mode collapse. Our approach is very different from these in combating mode collapse.

### 2.2 Conditional GANs

The most straightforward way to improve the performance of GANs is to incorporate side information of the samples, typically class labels. Depending on how the class information is used, the models are categorized into two types: projection-based models and classifier-based models.

Projection-based models [17] discriminate samples by projecting them onto two embeddings, meaning 'real' and 'fake', for the corresponding class and measuring the discrepancy of the projection values. As the network architectures are combined with attention modules [29] and increased to high-capacity [4], these models improve synthesizing large-scale and high-fidelity images, achieving the state-of-the-art performance. However, since they do not learn class embeddings through an explicit classifier, it may not be easy for the methods to learn the class-specific features.

In contrast, classifier-based models [21, 24] use an auxiliary classifier with a discriminator to explicitly learn class-specific features for generation. However, when incorporating the auxiliary classifier, the diversity of generated samples often degrades severely because the generator focuses on generating easily-classifiable samples [8, 30]. To prevent the generator from being over-confident to some samples, Zhou et al. [30] introduces an adversarial loss on the classifier. But, the loss often deteriorates the classifier and hinders the models from scaling up to large datasets (e.g., ImageNet). While Gong et al. [8] propose a scalable model using twin auxiliary classifiers, they require one more classifier to prevent the other from degeneration.

Unlike these methods, we address the mode-collapse issue based on the hypersphere-based discriminators by integrating only a single auxiliary classifier without any adversarial loss.

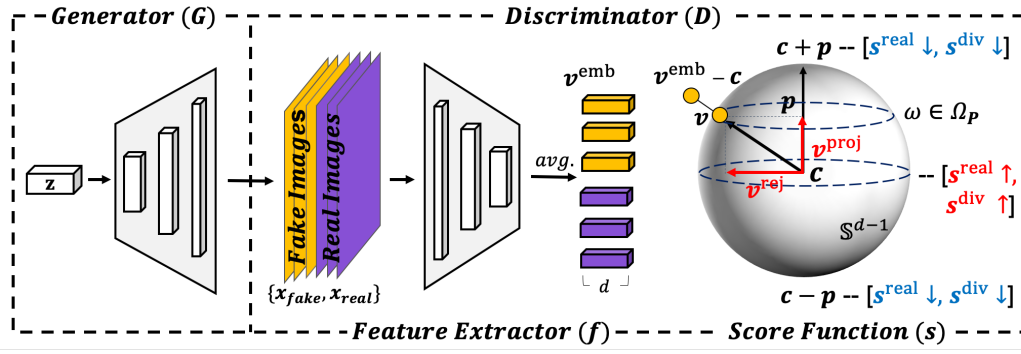

Figure 1: Overall architecture of CircleGAN. Given a randomly sampled noise vector $\boldsymbol{z}$, the generator synthesizes fake samples and the discriminator discriminates real samples from the fake samples. The discriminator produces the embeddings $\boldsymbol{v}^{\text{emb}}$, projects them onto the hypersphere by centering and $\ell 2$-normalization ($\boldsymbol{v}$), and discriminates them according to a score function based on their corresponding spherical circles. In training, CircleGAN performs adversarial learning based on the proposed score functions ($s^{\text{real}}$ and $s^{\text{div}}$). See text for details.

## 3    Proposed Approach

In the framework of GANs, our method, CircleGAN, learns a discriminator that projects samples on a hypersphere and then scores them using their corresponding spherical circles of the hypersphere. The overall architecture is illustrated in Fig. 1. The discriminator of CircleGAN is composed of feature extractor $f$ and score function $s$: $D(\boldsymbol{x}) = s(f(\boldsymbol{x})) = s(\boldsymbol{v})$. The key idea is to leverage the geometric characteristics of the hypersphere in scoring the quality of samples.

In the following, we first introduce CircleGAN for unconditional settings and then extend it to conditional settings. We also discuss our method in comparison to recent hypersphere-based method [22].

### 3.1    CircleGAN

Let samples $\mathcal{X} = \{\boldsymbol{x}\}_{i=1}^{n}$ be drawn from both data distribution $\mathcal{P}_{\text{data}}$ and the generator distribution $\mathcal{P}_{\text{gen}}$. The discriminator $D$ first embeds the samples $\mathcal{X}$ to $\mathcal{V}^{\text{emb}} = \{\boldsymbol{v}_i^{\text{emb}}\}_{i=1}^{n}$ and then projects them on a unit hypersphere with a learnable center $\boldsymbol{c}$: $\boldsymbol{v}_i = (\boldsymbol{v}_i^{\text{emb}} - \boldsymbol{c})/\|\boldsymbol{v}_i^{\text{emb}} - \boldsymbol{c}\|$. We denote the sets of sample embeddings on the hypersphere by $\mathcal{V} = \{\boldsymbol{v}_i\}_{i=1}^{n}$. Given a unit pivotal vector $\boldsymbol{p}$, which is learnable, we can define the set of spherical circles $\Omega_p$ that is perpendicular to $\boldsymbol{p}$. Then, each point $\boldsymbol{v}_i$ (i.e., sample embedding) on the hypersphere is assigned to a corresponding circle $\omega_{m(i)} \in \Omega_p$, where the function $m$ represents an injective mapping from $\mathcal{V}$ to $\Omega_p$. As shown in the right of Fig. 1, using the sample embedding $\boldsymbol{v}_i$ and the pivotal vector $\boldsymbol{p}$, we define its projected vector $\boldsymbol{v}_i^{\text{proj}}$ and its rejected vector $\boldsymbol{v}_i^{\text{rej}}$:

$$\boldsymbol{v}_i^{\text{proj}} = \langle \boldsymbol{v}_i, \boldsymbol{p} \rangle \boldsymbol{p}, \qquad \boldsymbol{v}_i^{\text{rej}} = \boldsymbol{v}_i - \boldsymbol{v}_i^{\text{proj}}, \tag{1}$$

where $\langle \cdot, \cdot \rangle$ indicates the inner product between two vectors. In a nutshell, each $\boldsymbol{v}_i$ corresponds to a spherical circle $\omega_{m(i)} \in \Omega_p$ that is identified with $\boldsymbol{v}_i^{\text{proj}}$. Note that both center $\boldsymbol{c}$ and pivotal vectors $\boldsymbol{p}$ are learned in training to adapt the hypersphere to the sample embeddings.

Our strategy in learning the discriminator is to populate real samples around the longest circle in $\Omega_p$, i.e., the great circle, while pushing fake samples toward the shortest circles in $\Omega_p$, i.e., the top or bottom pole. Since longer circles occupy the larger area on the hypersphere, they may allow more diversity in representation learning, and vice versa. In this sense, discriminating real and fake samples based on their corresponding circles can naturally induce diversity to generated samples.

We propose to measure the realness score for sample embedding $\boldsymbol{v}_i$ based on the proximity of its corresponding circle to the great circle, which is computed by

$$s^{\text{real}}(\boldsymbol{v}_i) = -\|\boldsymbol{v}_i^{\text{proj}}\|_2/\sigma^{\text{proj}}, \tag{2}$$

where the score is normalized with its standard deviation $\sigma^{\mathrm{proj}}$ to fix the scale consistently through the course of training and $\sigma^{\mathrm{proj}}$ is computed as $\sqrt{\sum_{i=1}^{|\mathcal{V}|}\|\boldsymbol{v}_i^{\mathrm{proj}}\|_2^2/|\mathcal{V}|}$. On the other hand, we define the diversifiability score for $\boldsymbol{v}_i$ by the radius of corresponding circle which can be quantified by

$$s^{\mathrm{div}}(\boldsymbol{v}_i) = \|\boldsymbol{v}_i^{\mathrm{rej}}\|_2/\sigma^{\mathrm{rej}}, \tag{3}$$

where $\sigma^{\mathrm{rej}}$ is computed as $\sqrt{\sum_{i=1}^{|\mathcal{V}|}\|\boldsymbol{v}_i^{\mathrm{rej}}\|_2^2/|\mathcal{V}|}$.

Note that the diversifiability score increases along with the realness score. Thus we use the realness score function $s^{\mathrm{real}}$ as the discriminator output so that it guides the generator in CircleGAN training. The discrimination based on the spherical circles increases the diversity of realistic samples while suppressing the diversity of unrealistic samples, which improves training the generator. This is supported by the experimental results in Sec. 4.2 and 4.3.

We design two other variants for scoring that explicitly combine both the realness score and the diversifiability score:

$$s^{\mathrm{add}}(\boldsymbol{v}_i) = -\|\boldsymbol{v}_i^{\mathrm{proj}}\|_2/\sigma^{\mathrm{proj}} + \|\boldsymbol{v}_i^{\mathrm{rej}}\|_2/\sigma^{\mathrm{rej}}, \tag{4a}$$

$$s^{\mathrm{mult}}(\boldsymbol{v}_i) = \arctan\left(\frac{\sigma^{\mathrm{proj}}}{\sigma^{\mathrm{rej}}} \frac{\|\boldsymbol{v}_i^{\mathrm{rej}}\|_2}{\|\boldsymbol{v}_i^{\mathrm{proj}}\|_2}\right). \tag{4b}$$

The former performs a simple addition of realness and diversifiability scores, and the latter measures an angle between the pivotal vector and sample embedding. The effects of these two variants will be demonstrated in our experiment.

To train the model based on the proposed score functions, we adopt the relativistic averaged loss [12] considering its robustness and simplicity:

$$\begin{aligned}\mathcal{L}_D^{\mathrm{adv}} = &- \mathbb{E}_{\boldsymbol{x}\sim\mathcal{P}_{\mathrm{data}}}\big[\log\big(\mathrm{sigmoid}\big(\tau(D(\boldsymbol{x}) - \mathbb{E}_{\boldsymbol{y}\sim\mathcal{P}_{\mathrm{gen}}}[D(\boldsymbol{y})])\big)\big)\big]\\ &- \mathbb{E}_{\boldsymbol{y}\sim\mathcal{P}_{\mathrm{gen}}}\big[\log\big(\mathrm{sigmoid}\big(\tau(\mathbb{E}_{\boldsymbol{x}\sim\mathcal{P}_{\mathrm{data}}}[D(\boldsymbol{x})] - D(\boldsymbol{y}))\big)\big)\big],\end{aligned} \tag{5}$$

where $\tau$ adjusts the range of score difference for the sigmoid functions and is set to 5 for $s^{\mathrm{add}}$ and $s^{\mathrm{real}}$ and 10 for $s^{\mathrm{mult}}$. For the adversarial loss of generator $\mathcal{L}_G^{\mathrm{adv}}$, we set it as the inverse of the discriminator loss by changing the source of the samples.

In the following, we introduce two additional losses to improve the training dynamics. The center estimation loss improves adapting hypersphere to the sample embeddings ($\boldsymbol{v}_i^{\mathrm{emb}}$) and the radius equalization loss increases discriminative power of the hypersphere by enforcing one-to-one mappings from the centered embeddings ($\boldsymbol{v}_i^{\mathrm{emb}} - \boldsymbol{c}$) to spherical embeddings ($\boldsymbol{v}$).

**Center estimation loss.** The center $\boldsymbol{c}$ is defined as a point that minimizes the sum of square of the distances to all the sample points. Thus, we optimize this directly by measuring the norms of the centered embeddings but proceed separately from the original objectives not to disrupt the adversarial learning. Here, we use Huber loss rather than $\ell 2$-loss.

$$\mathcal{L}_{\mathrm{center}} = \frac{1}{|\mathcal{V}|}\sum_{i=1}^{|\mathcal{V}|}\mathcal{L}_{\mathrm{huber}}(\|\boldsymbol{v}_i^{\mathrm{emb}} - \boldsymbol{c}\|_2), \tag{6}$$

where $\mathcal{L}_{\mathrm{huber}}$ is defined as $0.5x^2$ if $x \leq 1$, otherwise $x - 0.5$.

**Radius equalization loss.** We equalize the radius of centered embeddings through the discriminator. First, we compute the target radius $R$ by taking square root of averaged squared norm of the centered embeddings. Then, we penalize the difference in radiuses using Huber loss.

$$\mathcal{L}_D^{\mathrm{reg}} = \frac{1}{|\mathcal{V}|^2}\sum_{i=1}^{|\mathcal{V}|}\mathcal{L}_{\mathrm{huber}}(R - \|\boldsymbol{v}_i^{\mathrm{emb}} - \boldsymbol{c}\|_2), \tag{7}$$

where $R$ is computed by $\sqrt{\frac{1}{|\mathcal{V}|}\sum_{i=1}^{|\mathcal{V}|}\|\boldsymbol{v}_i - \boldsymbol{c}\|_2^2}$.

The total losses for the unconditional settings are $\mathcal{L}_D = \mathcal{L}_D^{\mathrm{adv}} + \mathcal{L}_D^{\mathrm{reg}}$ and $\mathcal{L}_G = \mathcal{L}_G^{\mathrm{adv}}$.

## 3.2 Extension to Conditional GANs

We extend our method for the conditional settings and improve the sample diversity of conditional GANs within each class. The key idea is to create multiple hyperspheres for target categories and perform adversarial learning in a class-wise manner. To be concrete, we create a center and a pivotal vector for each class: $\{\boldsymbol{c}\}_{l=1}^{L}$ and $\{\boldsymbol{p}\}_{l=1}^{L}$, where $L$ is the number of categories. The embeddings are translated with their corresponding center vectors, and the scores for adversarial learning are measured based on their corresponding pivotal vectors. For the rest, in computing non-trainable parameters (e.g., standard deviations in Eqs. 2, 3, 4 and target radius in Eq. 7) and comparing scores in the loss in Eq. 5, we take all samples and associate them together irrespective of class labels due to the limited batch sizes.

We incorporate an auxiliary classifier $C$ to learn the class-specific features. The auxiliary classifier is a simple linear layer attached to the discriminator and is trained without any adversarial loss to predict the class of a sample $\boldsymbol{x}$. With this classifier, the generator makes the sample class-specific by maximizing the probability of its corresponding label $y$:

$$\mathcal{L}_D^{\text{cls}} = \mathbb{E}_{(\boldsymbol{x},y)\sim\mathcal{P}_{\text{data}}}[-\log C(\boldsymbol{x})_y], \quad \mathcal{L}_G^{\text{cls}} = \mathbb{E}_{(\boldsymbol{x},y)\sim\mathcal{P}_{\text{gen}}}[-\log C(\boldsymbol{x})_y]. \tag{8}$$

The total losses for the conditional settings are $\mathcal{L}_D = \mathcal{L}_D^{\text{adv}} + \mathcal{L}_D^{\text{reg}} + \mathcal{L}_D^{\text{cls}}$ and $\mathcal{L}_G = \mathcal{L}_G^{\text{adv}} + \mathcal{L}_G^{\text{cls}}$. The overall algorithm with the proposed components is presented in Algorithm 1.

---

**Algorithm 1:** Training CircleGAN

**Given:** generator parameters $\psi_g$, discriminator parameters $\psi_d$, pivotal vectors $\{\boldsymbol{p}\}_{l=1}^{L}$, center vectors $\{\boldsymbol{c}\}_{l=1}^{L}$

**while** $\psi_g$ *not converged* **do**

    Compute embeddings $\mathcal{V}^{\text{emb}}, \mathcal{V}$ using $\mathcal{X}$ sampled from $\mathcal{P}_{\text{data}}$ and $\mathcal{P}_{\text{gen}}$

    Compute $\mathcal{L}_{\text{center}}$ by Eq. 6

    **Update center** $\boldsymbol{c}_l \leftarrow \text{Adam}(\nabla_{\boldsymbol{c}_l}\mathcal{L}^{\text{center}}, \boldsymbol{c}_l)$

    Compute realness $s^{\text{real}}(\boldsymbol{v})$ and diversifiability $s^{\text{div}}(\boldsymbol{v})$ by Eq. 2, 3

    Compute scores $s^{\text{real}}(\boldsymbol{v})$, $s^{\text{add}}(\boldsymbol{v})$ or $s^{\text{mult}}(\boldsymbol{v})$ by Eq. 2, 4

    Compute discriminator loss $\mathcal{L}_D$ by Eq. 5, 7, 8

    **Update discriminator and pivot** $[\psi_d, \mathbf{p}_l] \leftarrow \text{Adam}(\nabla_{\psi_d, \mathbf{p}_l}\mathcal{L}_D, [\psi_d, \mathbf{p}_l])$

    Compute embeddings $\mathcal{V}^{\text{emb}}, \mathcal{V}$ using $\mathcal{X}$ sampled from $\mathcal{P}_d$ and $\mathcal{P}_g$

    Compute realness $s^{\text{real}}(\boldsymbol{v})$ and diversifiability $s^{\text{div}}(\boldsymbol{v})$ by Eq. 2, 3

    Compute scores $s^{\text{real}}(\boldsymbol{v})$, $s^{\text{add}}(\boldsymbol{v})$ or $s^{\text{mult}}(\boldsymbol{v})$ by Eq. 2, 4

    Compute generator loss $\mathcal{L}_G$ by Eq. 5, 8

    **Update generator** $\psi_g \leftarrow \text{Adam}(\nabla_{\psi_g}\mathcal{L}_G, \psi_g)$

**end**

---

## 3.3 Comparison to SphereGAN

As ours, SphereGAN [22] also employs a hypersphere as the embedding space for the discriminator, but constructs it with a different projection function and learns with a different objective, as shown in Fig. 2. For hypersphere projection, CircleGAN uses translation and $\ell2$-normalization while SphereGAN uses inverse stereographic projection (ISP).

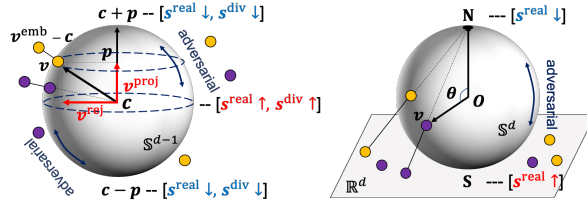

Figure 2: CircleGAN (ours) vs. SphereGAN [22].

The translation and $\ell2$-normalization tends to distribute samples evenly onto the hypersphere, but ISP concentrates a large portion of samples around the north pole and maps only a small portion around the south pole. This biased projection may prevent the discriminator from fully exploiting the space in learning. Furthermore, while CircleGAN learns the center and the pivotal point adaptively to sample embeddings, SphereGAN uses a fixed coordinate system fixed with the north pole $\mathbf{N}$ and the

origin **O**. As demonstrated in Sec. 4.2, our projection method shows better performance than ISP in practice. As for the objective in training, the score function $s_{\phi_i}^{\text{mult}}$ is analogous to the SphereGAN objective [22]; both maximize angles between a reference vector and sample embeddings. However, while CircleGAN maximizes the angles to the great circle, SphereGAN does it to the opposite point of the reference vector. The use of circles turns out to make a significant difference in sample diversity and quality, as shown in our experiments. Finally, CircleGAN easily extends to conditional settings by creating multiple hyperspheres, whereas SphereGAN is less flexible due to its specialized projection with the fixed reference vector on the hypersphere.

## 4 Experiments

We conduct experiments in both unconditional and conditional settings of GANs to demonstrate the effectiveness of the proposed methods. In all the experiments, we use the ResNet-based architecture for both the discriminator and the generator with some modifications from the original model [10]. The modifications and the training details are presented in the supplementary A. In Section 4.1, we describe the metrics to evaluate the realness and diversity of the generated samples. Then, in Section 4.2 and 4.3, we validate our approach by performing unconditional and conditional generation tasks on standard benchmark datasets. To further investigate the scalability of our model, we provide more results on the large-scale dataset, ImageNet, in the supplementary B.

### 4.1 Evaluation Metrics

The common evaluation metrics for image generation are Inception Score (IS) [24] and Frechet Incéption Distance (FID) [11]. IS measures how distinctively each sample is classified as a certain class and how similar the class distribution of generated samples is to that of the target dataset. FID measures a distance between the distribution of real data and that of generated samples in an embedding space, where the embeddings are assumed to be Gaussian distributed. While these are easy to calculate and correlate well with human assessment of generated samples, there have been some concerns about them. First, IS is computed based on class probabilities over Imagenet classes, and thus the evaluation on other datasets cannot be accurate since their class distributions are different from that of Imagenet [30]. Second, IS is highly sensitive to small changes in weights of classifiers [3] and image samples, as shown in our STL10 experiments of Sec. 4.2. Third, FID does not penalize a sample with a similar but different identity, e.g., a clear cat image generated by a dog label, which would be problematic particularly for conditional generations. Hence, in addition to IS and FID, we use two other metrics [25], GAN-train and GAN-test, for evaluation on conditional settings.

GAN-train and GAN-test evaluate diversity and quality of images, respectively. They can easily adapt to each target dataset and also consider mislabeled images in evaluation. GAN-train trains a classifier using generated images and then measures its accuracy on a validation set of the target dataset. This score is analogous to recall (the diversity of images) since the score would increase if the generated samples cover different modes of the target dataset. In contrast, GAN-test trains a classifier using a training set of the target dataset and then measures its accuracy on the generated images. This measure is not related to diversity, but to precision since high-quality samples even from a single mode can achieve a high score. To sum up, we evaluate the models of unconditional GANs with IS and FID, and along with these scores, we use GAN-train and GAN-test for conditional GANs. To measure IS and FID, we use 50K images for all experiments following original implementations.[1]

### 4.2 Unconditional GANs

For unconditional generation task, we evaluate our methods on CIFAR10 and STL10 [6]. CIFAR10 and STL10 consists of 50K $32 \times 32$ and 100K $96 \times 96$ images of 10 classes, respectively. We resize STL images to $48 \times 48$ before training, following the experimental protocol of [18, 22]. We compare our methods with two Lipschitz-based models [10, 18] and one hypersphere-based model [22] (Table 1a). The best and second-best results are highlighted with red and blue colors, respectively.

CircleGAN models achieve the best and second-best performance in terms of all metrics on the datasets, except for IS on STL10 where SphereGAN performs the best. We suspect that the exception

Table 1: Unnconditional GAN results on CIFAR10 and STL10.

(a) Comparison on CIFAR10 and STL10.

| Model | CIFAR10 | | STL10 | |
|---|---|---|---|---|
| | IS($\uparrow$) | FID($\downarrow$) | IS($\uparrow$) | FID($\downarrow$) |
| real images | 11.2 | 3.43 | 26.1 | 17.9 |
| WGAN-GP [10] | 7.76 | 22.2 | 9.06 | 42.6 |
| SNGAN [18] | 8.22 | 21.7 | 9.10 | 40.1 |
| SphereGAN [22] | 8.39 | 17.1 | **9.55** | 31.4 |
| CircleGAN ($s^{\text{real}}$) | **8.54** | **12.2** | 9.18 | **27.0** |
| CircleGAN ($s^{\text{add}}$) | **8.55** | **12.3** | **9.24** | **27.5** |
| CircleGAN ($s^{\text{mult}}$) | 8.47 | 12.9 | 8.82 | 30.1 |

(b) Ablation study on CIFAR10.

| Methods | FID($\downarrow$) |
|---|---|
| CircleGAN ($s^{\text{mult}}$) | 12.9 |
| - radius equalization | 14.6 |
| - center estimation | 15.2 |
| - circle learning | 15.8 |
| - score normalization | 16.8 |
| - $\ell$2-projection | 20.4 |

is due to the sensitivity of IS, as also reported in [3]; IS on the test set of STL10 is 14.8, which is significantly lower than 26.1 on the training set, but for other datasets such as CIFAR10 and CIFAR100, IS values are similar between train and test sets (CIFAR10: 11.2 vs. 11.3, CIFAR100: 14.8 vs. 14.7).

To further compare CircleGAN to SphereGAN, we conduct ablation studies on CIFAR10 with the model using angle $s^{\text{mult}}$ for score function (Table 1b). Each component in the table is subsequently removed from the full CircleGAN model to see its effect. Here we use the FID metric, which is more stable. First and second, we remove radius equalization and center estimation losses, respectively. Third, we replace the CircleGAN objective with that of SphereGAN, which maximizes the angle to the opposite of pivotal point. Forth, we remove the score normalization. Fifth, we replace $\ell$2-normalization with ISP of SphereGAN. The results show that the proposed components consistently improve FIDs. In particular, replacing $\ell$2-normalization with ISP significantly deteriorates FID, which implies that ISP of SphereGAN may be problematic due to the embedding bias of samples. CircleGAN not only outperforms SphereGAN with a large margin, but also easily extends to conditional settings as demonstrated in the next experiment.

## 4.3 Conditional GANs

We conduct conditional generation experiments on CIFAR10, CIFAR100 [15] and TinyImagenet.[2] CIFAR100 consists of 50K $32 \times 32$ images of 100 classes and TinyImageNet consists of 100K $64 \times 64$ images of 200 classes. We compare our models with a projection-based model [17] and also with two classifier-based models [30, 21], one with adversarial losses on class probability [30] and the other without the losses [21]. We present the results in Table 2, 3, 4 for CIFAR10, CIFAR100, and TinyImagenet, respectively. The numbers inside the parentheses indicate the results of our models without the auxiliary classifier $C$. All the CircleGAN models outperform the other models in terms of all metrics across all the datasets. On the datasets with more classes and more diverse samples with higher resolution, the performance gains over other models become greater. It demonstrates the advantage of class-wise hypersphere placing realistic samples around the great circle in a class-wise manner.

Table 2: Conditional GAN results on CIFAR10.

| Model | IS($\uparrow$) | FID($\downarrow$) | GAN-train($\uparrow$) | GAN-test($\uparrow$) |
|---|---|---|---|---|
| real images | 11.2 | 3.43 | 92.8 | 100 |
| AC+WGAN-GP [10] | 8.27 | 13.7 | 79.5 | 85.0 |
| Proj. SNGAN [17] | 8.47 | 10.4 | 82.2 | 87.3 |
| AMGAN [30] | 8.79 | 7.62 | 81.0 | 94.5 |
| CircleGAN ($s^{\text{real}}$) | **9.08** (8.91) | **5.72** (7.47) | **87.0** (84.0) | 96.1 (84.5) |
| CircleGAN ($s^{\text{add}}$) | 9.01 (8.80) | 5.90 (8.09) | **86.8** (82.6) | **96.6** (82.9) |
| CircleGAN ($s^{\text{mult}}$) | **9.22** (8.83) | **5.83** (12.2) | 86.3 (83.5) | **96.8** (83.5) |

Table 3: Conditional GAN results on CIFAR100.

| Model | IS($\uparrow$) | FID($\downarrow$) | GAN-train($\uparrow$) | GAN-test($\uparrow$) |
|---|---|---|---|---|
| real images | 14.8 | 3.92 | 69.4 | 100.0 |
| AC+WGAN-GP [10] | 9.10 | 15.6 | 26.7 | 40.4 |
| Proj. SNGAN [17] | 9.30 | 15.6 | 45.0 | 59.4 |
| AMGAN [30] | 10.2 | 16.5 | 23.2 | 70.8 |
| CircleGAN ($s^{\text{real}}$) | 11.8 (9.93) | **7.43** (9.45) | **54.7** (48.6) | **93.9** (58.5) |
| CircleGAN ($s^{\text{add}}$) | **11.9** (10.13) | **7.35** (8.99) | **55.6** (49.9) | **92.5** (57.7) |
| CircleGAN ($s^{\text{mult}}$) | **11.9** (9.98) | 8.62 (9.10) | 54.0 (47.4) | 91.0 (58.0) |

The GAN-train and GAN-test results show that CircleGAN produces higher quality and more diverse samples than other models. In particular for the GAN-test, it remains almost the same at the highest level regardless of the datasets, which implies that every sample captures important features of its corresponding class to be classified correctly by the pre-trained classifier. We attribute this to the auxiliary classifier since all the scores drop significantly on all datasets when the classifier is detached. The overall scores without auxiliary classifier are similar to the projection-based models, which have no classifier. Hence, this supports the use of auxiliary classifier for learning the class-specific features in discriminator and correspondingly to update samples in generator.

However, the auxiliary classifier does not benefit all the other classifier-based models [10, 30], instead they perform significantly worse as the number of classes or the resolution increases. Specifically, AMGAN [30], whose discriminator is trained with an adversarial loss on class probability, shows decent performance on CIFAR10, but dramatically degrades on CIFAR100 and fails to train on TinyImagenet. The other classifier-based model having no adversarial loss also [10, 21] shows competitive results on CIFAR10 and CIFAR100, but poor GAN-train and GAN-test scores on TinyImagenet. These results suggest that utilizing spherical circles is highly effective and flexible to integrate the auxiliary classifier in conditional GANs.

Table 4: Conditional GAN results on TinyImagenet.

| Model | IS($\uparrow$) | FID($\downarrow$) | GAN-train($\uparrow$) | GAN-test($\uparrow$) |
|---|---|---|---|---|
| real images | 33.3 | 4.59 | 59.2 | 100.0 |
| AC+WGAN-GP [10] | 10.3 | 32.5 | 0.4 | 0.7 |
| Proj. SNGAN [17] | 9.38 | 42.0 | 22.6 | 19.3 |
| CircleGAN ($s^{\text{real}}$) | **21.6** | **15.5** | **30.4** | **94.6** |
| CircleGAN ($s^{\text{add}}$) | **20.8** | **15.6** | 28.8 | **94.9** |
| CircleGAN ($s^{\text{mult}}$) | 20.8 | 17.5 | **30.4** | 93.4 |

For qualitative evaluation, we sample images and obtain t-SNE using pre-trained classifier for CircleGAN and Proj. SNGAN [17], which is the most competitive algorithm on TinyImagenet. We use 5 classes of images where the classes are selected by the author not to be overlapped in nature: goldfish, yorkshire-terrier, academic gown, birdhouse, schoolbus (Fig. 3). The results demonstrate that the images synthesized from CircleGAN correspond to their classes and almost overlapped to the train and the validation sets of the dataset at the 2D t-SNE space, but the images from Proj. SNGAN are vague in perceiving them as instances of their corresponding classes. Also, the distribution of t-SNE itself certifies that the samples from Proj. SNGAN are far apart from the support of the dataset. The more qualitative results can be found in supplementary material C.

# 5 Conclusion

In this paper, we have demonstrated that learning and discriminating sample embeddings using their corresponding spherical circles on a hypersphere is highly effective in generating diverse samples of high quality. The proposed method provides the state-of-the-art performance in unconditional generation and also extends to conditional setups with class labels by creating hyperspheres for the classes. The impressive performance gain over the recent methods on standard benchmarks demonstrates the effectiveness of the proposed approach.

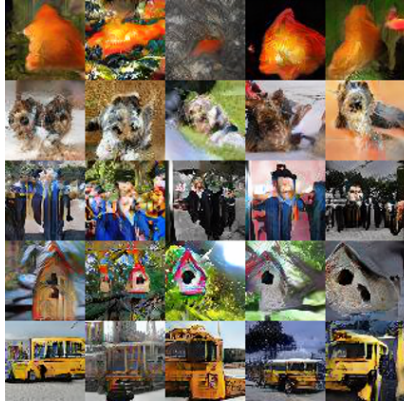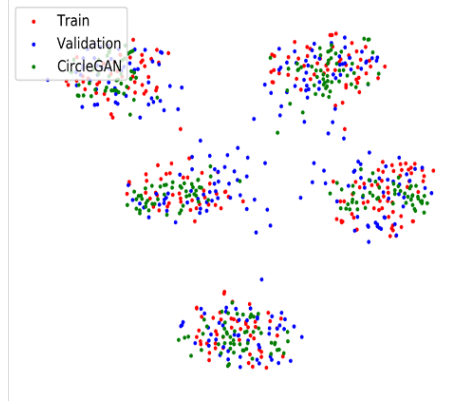

(a) CircleGAN (ours) images and embeddings

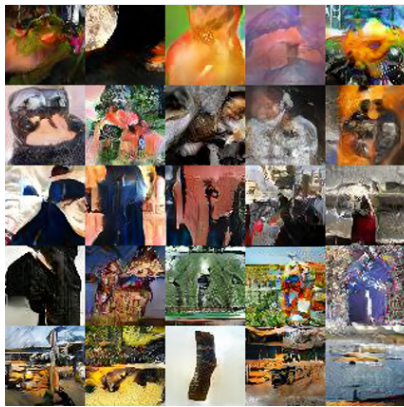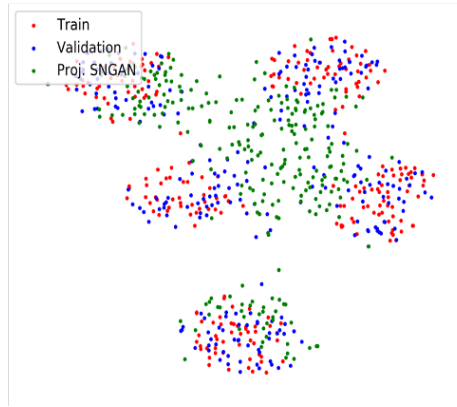

(b) Proj. SNGAN [17] images and embeddings

Figure 3: CircleGAN vs. Projection-SNGAN on TinyImagenet. For 5 classes (goldfish, yorkshire-terrier, academic gown, birdhouse, schoolbus), generated images and their 2D embeddings from t-SNE are visualized. For t-SNE, we train a classifier using the training set and use it for embedding the generated images. For comparison, we also use 250 images randomly taken from train and validation sets, respectively.

## Broader Impact

This work addresses the problem of generative modeling and adversarial learning, which is a crucial topic in machine learning and artificial intelligence; b) the proposed technique is generic and does not have any direct negative impact on society; c) the proposed model improves sample diversity, thus contributing to reducing biases in generated data samples.

## Acknowledgments

This research was supported by Basic Science Research Program (NRF-2017R1E1A1A01077999) and Next-Generation Information Computing Development Program (NRF-2017M3C4A7069369), through the National Research Foundation of Korea (NRF) funded by the Ministry of Science and ICT (MSIT), and also by Institute for Information & communications Technology Promotion (IITP) grant funded by the Korea government (MSIP) (No. 2019-0-01906, Artificial Intelligence Graduate School Program (POSTECH)).

## Footnotes

[1]IS: https://github.com/openai/improved-gan, FID: https://github.com/bioinf-jku/TTUR

[2]https://tiny-imagenet.herokuapp.com/

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
