[Supplementary Material]

# Supplementary Material for CircleGAN

**Woohyeon Shim**
POSTECH CiTE
wh.shim@postech.ac.kr

**Minsu Cho**
POSTECH CSE & GSAI
mscho@postech.ac.kr

## 1  Implementation Details

We provide hyperparameter settings and architectural details used in our work.

**Hyperparameter.**  All the experiments use the same hyperparameters. We train both the discriminator and the generator using ADAM optimizer [5] with $\beta_1 = 0.5$, $\beta_2 = 0.999$, initial learning rate = 0.0001, minibatch size = 64, and total of 300K iterations each with 1:1 balanced schedule. We set the total iterations to be 3 times of the iterations of SN- and GP- based models (100K), because these models update the network with 1:5 schedule.

**Network architecture.**  All the models of unconditional and conditional CircleGANs use Resnet-based architectures [1, 3, 7]. Here, we change some regularizers and tricks to the techniques adopted in DCGANs and other classifier-based cGANs [8, 12]. The major differences are as follows: 1) we use dropout and BN with weight normalization (WN) as a regularizer instead of the existing techniques such as spectral normalization (SN) and gradient penalties (GP). The BN is proven to function as a regularizer imposing the Lipschitz constraint [9], which has been achieved by SN and GP [3, 7]. Plus, the dropout and WN have been successfully adopted in the classifier-based model [12]. 2) we feed the fake and real samples together into the discriminator to mimic the target distribution directly, not the whitened one normalized by the BN layers.

We provide the architectural details for the unconditional and the conditional CircleGANs, where we borrow some expressions from [10]. We denote the ResNet blocks (see Fig. 1) with 1) $R_d^S$, $R_d^D$ and $R_d^U$ for the blocks which produce the same resolution, downsampled and upsampled outputs by a factor of 2, respectively, 2) $R_d^{D,1st}$ for the first block in the discriminator, where $d$ denotes the channel dimensions. Also, we denote with 1) $D_k$ a linear layer with $k$ dimensions, 2) $G$ a global average pooling layer, 3) $T_h$ a layer that transposes the input feature map to have the target resolution $h \times h$ and 4) $C_3^S$ a block that outputs the image of the same resolution as the input, which consists of BN, Relu, Conv(3), and tanh layers. The architecture configuration on both the unconditional and the conditional settings are shown for each dataset in Table 1, 2.

(a) $R_d^S$, $R_d^D$ and $R_d^U$      (b) $R_d^{D,1st}$

Figure 1: ResNet blocks used in (a) the discriminator and the generator and (b) the first block in the discriminator. We distinguish the blocks producing the different resolution of output from the input with a dashed line in (a).

Table 1: The CircleGAN architectures for unconditional settings.

| Dataset | Generator - Discriminator |
|---|---|
| CIFAR10 | $\dfrac{D_{4096}\text{-}P_4\text{-}R^U_{256}\text{-}R^U_{256}\text{-}R^U_{256}\text{-}C^S_3}{R^{D,1st}_{128}\text{-}R^D_{128}\text{-}R^D_{128}\text{-}R^S_{128}\text{-}G}$ |
| STL10 | $\dfrac{D_{9216}\text{-}P_6\text{-}R^U_{256}\text{-}R^U_{256}\text{-}R^U_{256}\text{-}C^S_3}{R^{D,1st}_{256}\text{-}R^D_{256}\text{-}R^D_{256}\text{-}R^S_{256}\text{-}G}$ |

Table 2: The CircleGAN architectures for conditional settings.

| Dataset | Generator - Discriminator |
|---|---|
| CIFAR10 | $\dfrac{D_{4096}\text{-}P_4\text{-}R^U_{256}\text{-}R^U_{256}\text{-}R^U_{256}\text{-}C^S_3}{R^{D,1st}_{128}\text{-}R^D_{128}\text{-}R^D_{128}\text{-}R^S_{128}\text{-}G\text{-}D_{10}}$ |
| CIFAR100 | $\dfrac{D_{4096}\text{-}P_4\text{-}R^U_{256}\text{-}R^U_{256}\text{-}R^U_{256}\text{-}C^S_3}{R^{D,1st}_{64}\text{-}R^D_{128}\text{-}R^D_{256}\text{-}R^S_{512}\text{-}G\text{-}D_{100}}$ |
| TinyImageNet | $\dfrac{D_{4096}\text{-}P_4\text{-}R^U_{256}\text{-}R^U_{256}\text{-}R^U_{256}\text{-}R^U_{256}\text{-}C^S_3}{R^{D,1st}_{64}\text{-}R^D_{128}\text{-}R^D_{256}\text{-}R^D_{512}\text{-}R^S_{1024}\text{-}G\text{-}D_{200}}$ |
| ImageNet | $\dfrac{D_{16384}\text{-}P_4\text{-}R^U_{1024}\text{-}R^U_{512}\text{-}R^U_{256}\text{-}R^U_{128}\text{-}R^U_{64}\text{-}C^S_3}{R^{D,1st}_{64}\text{-}R^D_{128}\text{-}R^D_{256}\text{-}R^D_{512}\text{-}R^D_{1024}\text{-}R^S_{1024}\text{-}G\text{-}D_{1000}}$ |

## 2  ImageNet Experiments

We present additional results of high-resolution image generation using ImageNet [2] with $128 \times 128$ resolution, which consists of 1.3M images of 1000 classes. We use the same hyperparameter settings and the network configuration used in other datasets, except the learning rates and the number of layers and filters in the networks. The learning rates of the discriminator and the generator are set according to two-timescale learning rate (TTUR) [4], which is adopted in Proj. SNGAN [6]. Proj. SNGAN sets the learning rates of the discriminator and the generator as 0.0004 and 0.0001, respectively, and they are fixed over the course of the training. Using this as our basic settings, we run an additional experiment where the learning rate of the discriminator is set to $8\times$ of the learning rate of the generator. The architectural details for ImageNet are presented in Table 2.

The quantitative results are shown in Fig. 2. We use CircleGAN - $s^{\mathrm{mult}}$ as our basic model. Starting from the fairly good performances in terms of both IS and FID, CircleGAN outperforms the best performance of Proj. SNGAN (Fig. 2, blue line) with significantly fewer iterations by a large margin (Fig. 2, orange and gray lines). However, our model undergoes complete training collapse as BigGAN does at a performance similar to ours [1]. To combat the collapse, we simply decay the learning rate of the generator linearly to 0 during 300K iterations (Fig. 2, yellow line), reaching another significant

(a) Learning curve of IS

(b) Learning curve of FID

Figure 2: Comparison of IS and FID on Imagenet with Proj. SNGAN [6]. IS: higher is better. FID: lower is better.

Table 3: Comparison of IS and FID on Imagenet with other state-of-the-art algorithms in conditional settings. IS: higher is better. FID: lower is better.

| Model | IS | FID |
|---|---|---|
| Proj. SNGAN [6] | 36.80 | 27.62 |
| SAGAN [11] | 52.52 | 18.65 |
| BigGAN [1] | 98.76 | **8.73** |
| CircleGAN - $s^{\mathrm{mult}}$ | **156.57** | 22.34 |

gain in performance. We provide a comparison of our algorithm with other state-of-the-art algorithms in Table 3.

For qualitative results, we visualize the images sampled from the same categories in the TinyImagenet (goldfish, yorkshire-terrier, academic-gown, birdhouse, schoolbus) in Fig. 3. Despite the remarkable performance of CircleGAN, there are still opportunities for further enhancements; class-specific features are well-preserved to each image, but monotonous are the images and look similar to each other. Training with larger batch size ($256 \rightarrow 2048$) and its relevant techniques [1] can help to address this issue, but we leave it to the future work.

(a) Goldfish

(b) Yorkshire-terrier

(c) Academic gown

(d) Birdhouse

(e) Schoolbus

Figure 3: ImageNet images randomly sampled from CircleGAN - $s^{\mathrm{mult}}$ for 5 classes (goldfish, yorkshire-terrier, academic gown, birdhouse, schoolbus).

# 3 Additional Results

In this section, we show qualitative results on CIFAR10 and STL10 for unconditional settings and CIFAR10, CIFAR100 and TinyImagenet for conditional settings (Fig. 4, 5).

(a) (Unconditional) CIFAR10 Images.

(b) (Unconditional) STL10 Images.

(c) (Conditional) CIFAR10 Images.

(d) (Conditional) CIFAR100 Images.

Figure 4: CIFAR10, STL10, CIFAR100 images randomly sampled from unconditional and conditional CircleGANs - $s^{\text{mult}}$.

Figure 5: TinyImagenet images of 25 randomly sampled classes from conditional CircleGAN - $s^{\mathrm{mult}}$.