[Reviews · NeurIPS 2020]

Review 1

Summary and Contributions: This paper present a new score function on a hypersphere to evaluate both realness and diversity of samples, which use the hypersphere as an embedding space and make spherical circles share the same normal vector as isolines, assume the real samples achieve the maximum realness and diversity, and reside in the great circlethe of hypersphere. The final experimental result is amazing.

Strengths: 1. The method improves the sample diversity along with the quality by adopting a hypersphere as an embedding space and performing the adversarial game on the most diversifiable region in the hypersphere. 2. Using the great circle on a hypersphere to place embeddings of the real samples on the most diversifiable region, rather than a point-based in conventional GANs.

Weaknesses: 1. The author does not give a explanation why the real samples that reside in the great circle indicates the maximum diversity. Does it mean that the diversity of generation samples is lower than the original image set? 2. The key information is missing in the description of the propoed method in Section 3, the statement is incoherent, and some symbols in the formulas (such as 'V_i', 'V^proj', 'p^~' in equal (1), 'S_r' in equal (5)) are not given specific meaning and explanation. So it is hard to read. 3. It is suggested to add a hypersphere chart to reflect the true intention of author, like 'make spherical circles sharing the same normal vector as isolines'.

Correctness: Intuitively, the hyperplane approach might be more effective. But this paper is hard to read and I am confused about the method described by the author. Although the result of the comparative experiment is amazing, I am not quite convinced.

Clarity: The article is poorly written and is not fully expressed, especially in the key of method description section.

Relation to Prior Work: Lack of discussion of existing methods, especially the Spherical GAN which also has the idea of hypersphere.

Reproducibility: No

Additional Feedback: 1. It may need to rearrange the text of the method section, and make clear the meaning of all custom symbols. 2. It need a further discussion about existing GAN methods which also use hypersphere, to batter present the characteristics of the propoesd method.


Review 2

Summary and Contributions: The authors incrementally suggest the novel method from baseline method which implements the point-based evaluation of realness on hyper-sphere. Contrary to baseline method, they aggressively searches the optimal distance by proposing following techniques: updating center point, incorporating various type of techniques including center estimation and radius equalization. Due to the such well-posed techniques, proposed method produces better results. The authors suggest new type of GAN objective functions for conditional GANs and shows the promising results.

Strengths: The authors have developed novel approaches to measure the discrepancy on hyper-sphere with various techniques. The experiments section is carefully composed. The proposed method outperforms state-of-the-art methods including baseline.

Weaknesses: In recent theoretical-oriented GAN models, they focus on suggesting new type of metrics between probability measures, and claim their theoretical superiority including stability, diversity in terms of probabilistic point of view. Contrary to recent trends, the proposed method starts from the combination of heuristic motivations and incrementally propose new objective function for GANs. The proposed method is some what interesting, but it is unreliable due to the lack of theoretical analysis.

Correctness: I like the ideas and concepts of 'diversity' and 'realness' on the sphere (which is projected by simple L2-normalization), but it is non-trivial to say that proposed objective function actually minimizes some 'distance' between real and fake probability distribution. SphereGAN implements IPMs as their objective function and shows the equivalence relation between minimizing Wasserstein distance in hyper-sphere and minimizing objective functions, but this kind of analysis is not dealt in proposed method even if SphereGAN is main baseline method. Thus authors needs to clarify what to minimize. The proposed method uses L2-normalization as a projection onto hyper-sphere which induces information loss as it is not one-to-one (All the conventional features lying in same lay started at origin is projected to same point in hyper-sphere). The stereo-graphic projection not only admits single fixed point where north pole ('center' in the paper) can be rotated transitively on the hyper-sphere. In this point of view, i think stereo-graphic projection can be applied to proposed method by replacing L2-normalization. Analysis of learning dynamic have not been dealt in the paper while this is main issue of GAN objective function. The stability of GAN learning have been investigated from the WGAN to SobolevGAN, in terms of designing gradient penalty. At least, the authors need to show the FID (or loss) landscape of proposed method. Overall, I'm not sure what brings the performance improvement in proposed method compared to baseline methods.

Clarity: Overall, I found this paper to be a nice read. The paper is well-written and structured clearly.

Relation to Prior Work: The authors properly discussed the difference between their method and baseline method.

Reproducibility: Yes

Additional Feedback:


Review 3

Summary and Contributions: The paper proposes a new GAN model that replace the original binary latent discrimination of the vanilla GAN by using a hypersphere space. This allows a "discrimination" on the quality and diversity of the input. The model can be extended to a conditional-GAN framework. Experiments are conducted on various image generation tasks, demonstrating the advantages of the proposed model.

Strengths: Overall, I think this is a very good paper. While the projecting (to the hypersphere) part is similar to SphereGAN, the paper present a new criterion on the discriminator, which considers both the realness and the diversity. These seems to work well with the hypersphere setup. The experiments compare the proposed CircleGAN with other GANs, where CircleGAN provides better overall performance. Ablation studies over SphereGAN is also included, which makes the proposal more convincing.

Weaknesses: 1) One question I have is on the computation cost of the proposed model. Since CircleGAN has additional training steps, and needs to update center, disc, and pivot, what is its training speed when comparing to, say SphereGAN? 2) In table 1(b), it seems without great circle learning (model 4), the model can still achieve good performance, even better than model 1 (IS). I'm wondering why is that?

Correctness: I think overall the paper is theoretically sound

Clarity: The paper is clearly written and easy to follow.

Relation to Prior Work: The proposed circleGAN is motivated from previous SphereGAN, which also use hypersphere embedding space. The proposed method add additional criterion for evaluation the realness and diversity of the generated samples, which are demonstrated to be beneficial for improving the performance of GAN.

Reproducibility: Yes

Additional Feedback:


Review 4

Summary and Contributions: this paper improve the sample diversity along with the quality by adopting a hypersphere as an embedding space and performing the adversarial game on the most diversifiable region in the hypersphere

Strengths: the idea is new, as far as I know, no similar idea was ever proposed in GAN framework.

Weaknesses: 1. It is recommended utilise a figure to show a brief illustration of the core idea for easy understanding. 2. lack experiment on large dataset and big image, e.g. imagenet. 3. The generated images seem not so good, and is far from recent sota GAN models such as stargan, stylegan. It will be better if the proposed idea was evluated on a stronger GAN baseline.

Correctness: yes

Clarity: yes

Relation to Prior Work: yes

Reproducibility: Yes

Additional Feedback: please see the weaknesses post rebuttal: after reading the rebuttal, I keep my original reate

[Author Response · NeurIPS 2020]

We thank all reviewers for their constructive feedback. Below are our responses to the concerns raised by the reviewers.
Although we cannot respond to all the comments, we will do our best to reflect all of them in our final manuscript.

**[R1] Assumptions on CircleGAN.** We assume the real samples have maximum diversity and quality, and those
objectives are achieved at the largest isoline, which is the great circle. Thus, we place the real samples to the great
circle, which indicates the diversity of generated samples is lower than the real samples.

**[R1, R4] Comparison between CircleGAN (ours) and SphereGAN [22].** As R1 and R4 suggested, we created Fig. 1
below, where overall architecture and symbols are described and relevant concepts in training are compared between the
two models. The superscripts ('emb', 'cen', 'proj', 'rej', 'real', 'div') show the status of a variable, and the subscripts
('$i$', '$r_i$', '$f_i$') denote the index or source of a variable. If clear from the context, we often omit the subscript $i$. For
notational simplicity, we omit tilde from $\tilde{p}$ to denote the unit pivotal vector.

(a) Overall Architecture

(b) CircleGAN Training (left) vs. SphereGAN Training (right)

Figure 1: CircleGAN (ours) vs. SphereGAN [22]. (a) Given a randomly sampled noise vector $z$, the generator
synthesizes fake samples and the discriminator produces the embeddings $v^{\text{emb}}$ from fake and real samples. (b)
CircleGAN projects these embeddings onto the hypersphere by centering ($v^{\text{cen}}$) and $\ell 2$-normalization ($v$), whereas
SphereGAN performs inverse stereographic projection. Due to learnable pivotal $p$ and center $c$ vectors, CircleGAN is
more flexible and amenable to conditional settings. Note, however, that in SphereGAN they are fixed to the north pole
$N$ and the origin $O$, respectively. In training, CircleGAN performs adversarial learning based on the great circle using
the proposed score functions ($s^{\text{real}}$ and $s^{\text{div}}$), and SphereGAN performs based on the point $N$, which causes lack of
sample diversity. [R1: The isolines of spherical circles in CircleGAN are delineated with dashed lines.]

**[R2] Theoretical analysis and comparison of projection methods.** If we use the Wasserstein distance and replace
the cost function (Eq. 8 of [22]) with our proposed score functions (Eq. 3, 4 of ours), all the theoretical results of
SphereGAN should also hold for CircleGAN. In our experiments, however, we use the relativistic objective of Eq. 5
because the work of [11] has shown that the Wasserstein objective is a specific form of relativistic objectives and the
form of Eq.5 produces superior performance than the Wasserstein objective. We think similar effects of the theoretical
results in SphereGAN may also hold for our case because: (1) the projection function has nothing to do with the cost
function of the Wasserstein distance and thus does not change theoretical results of SphereGAN, (2) both CircleGAN
and SphereGAN use the same form of hypersphere embedding space, (3) the cost function is almost the same to the
angles (or $\ell 2$-distances) from each particular reference, satisfying the conditions (non-negative & bounded) for the
Wasserstein space to be defined and allowing propositions 1 and 2 in SphereGAN to be applied, and (4) propositions 1
and 2 state that the convergence in the Wasserstein space is equivalent to that of the Wasserstein objective.

The superiority of CircleGAN mainly comes from learning based on the great circle. The inverse stereographic
projection (ISP) is problematic for this approach because ISP results in a great circle with a lower sample density
than the north pole and thus prevents the embedding space from representing the sample diversity. In this sense,
$\ell 2$-normalization is adequate since it allows the great circle to have the highest sample density w.r.t. the pivotal point.

**[R3] Effectiveness of learning based on the great circle.** We conjecture the setting of [Table 1b, (4)] without other
techniques diminishes the efficacy of learning based on the great circle. To support this assumption, we directly ablate
the great circle based learning from the final models for both unconditional and conditional settings on CIFAR10. The
results worsen significantly on both unconditional settings (IS: 8.47 -> 8.57, FID: 12.9 -> 32.7) and conditional settings
(IS: 9.22 -> 8.41, FID: 5.83 -> 19.0). These results validate utilizing the great circle for training GANs.

**[R3] Additional computational cost.** For each iteration of the discriminator update, both center and pivotal vectors
are jointly updated using a control dependency. Thus, the increase in computational cost over the baseline is negligible.

**[R2, R4] Learning dynamics and Imagenet experiments.** In supplementary materials (L27-L49), we provide the
experiments on a large-scale Imagenet dataset with $128 \times 128$ resolution along with the learning curve of IS and FID.
CircleGAN outperforms recent methods [4, 18, 29] by a large margin using significantly fewer iterations. We expect the
stability and qualitative results to further improve by increasing the batch size as do other state-of-the-art methods.

[Meta-Review · NeurIPS 2020]

This paper proposes a new GAN training technique based on intuition about the hypersphere. It attains state-of-the-art IS and FID scores on a few datasets. Reviewers were initially confused and concerned about its similarity to SphereGAN but were convinced it should be accepted after the rebuttal.